# The Eternal Struggle for the City: In Search of an Alternative Framework for Citizen Participation in Urban Regeneration Projects in Shrinking Cities

Maja Ročak [1,2,3,*] and Sabrina Keinemans [1]

1   Research Centre for Social Integration, Zuyd University of Applied Sciences, Ligne 1,
    6131 MT Sittard, The Netherlands; sabrina.keinemans@zuyd.nl
2   NEIMED, Socio-Economic Knowledge Institute, Valkenburgerweg 177, 6419 AT Heerlen, The Netherlands
3   Research Centre for Social Resilience, Fontys University of Applied Sciences, Dominee Theodor
    Fliednerstraat 2, 5612 MA Eindhoven, The Netherlands
*   Correspondence: maja.rocak@zuyd.nl

**Abstract:** The relevance of citizen participation in regeneration projects, particularly in shrinking cities, is widely acknowledged, and this topic has received a great deal of policy and academic attention. Although the many advantages of citizen participation in regeneration projects have been identified, its current forms have also received considerable criticism. In short, this criticism boils down to the conclusion that the ideal of citizen participation is not put into practice. This paper considers why this is the case, asking whether current participatory practices enable citizens to exercise influence as political actors in urban regeneration projects. In this paper, we examine this question based on Mouffe's conception of the political, coupled with findings from our empirical research conducted in Heerlen North, The Netherlands. We conducted qualitative research on urban regeneration in the shrinking old industrial city of Heerlen. The findings reveal two distinct perspectives on citizen participation. Professionals see the existing context of citizen participation as a reasonable and practical but, in some respects, insufficient practice. Citizens' views on participation are organized around feelings of anger, shame, and fear and are grounded in experiences of a lack of recognition. These experiences limit citizens' abilities to exert true influence on regeneration projects. We conclude that efforts to regenerate shrinking cities should strive to recognize these experiences so as to create conditions that generate respect and esteem and, as such, enable urban social justice.

**Keywords:** citizen participation; shrinking cities; urban transformation; regeneration projects; agonistic space; territorial stigmatization; recognition; urban social justice





## 1. Introduction

Areas faced with socioeconomic deprivation and territorial stigmatization, especially in shrinking cities, have continually attracted the attention of politicians, policy makers, and planners [1–3]. Urban regeneration projects, which are often far-reaching in terms of socio-spatial interventions, have been common in Europe since roughly the 1970s. These projects are common practice in shrinking cities, urban areas faced with continuous or temporary population and economic decline [2] and territorial stigmatization [4]. Although deprived districts can be found in both growing and shrinking cities, in shrinking cities, many of the issues targeted by regeneration are exacerbated by the context of shrinkage [5]. It is in these neighborhoods that the causes and outcomes of shrinkage mutually reinforce one another, leading to regeneration projects in these areas. In The Netherlands, urban regeneration projects are guided by the central government and have spread to many of the country's deprived districts. Some recent well-known examples in The Netherlands include Rotterdam Zuid, Heerlen Noord, and Woensel Zuid. Most notably, many regeneration projects are taking place in the shrinking city of Heerlen.

Many of the districts targeted by regeneration projects face various kinds of disadvantage (including the out-migration of prosperous young people, an ageing population, lower socioeconomic status, poor (mostly social) housing stock, inadequate infrastructure, and poor environmental standards), as well as territorial stigmatization, that is, a collective representation of a place that contributes to the social reproduction of inequality and marginality [6]. Regeneration projects often strive for so-called "levelling up", aiming to reach many goals simultaneously in order to improve the living situation in these districts [7]. Currently, this holistic orientation is, in the Dutch context, expressed through the concept of "brede welvaart" or "widespread prosperity". In this respect, regeneration projects address not only physical and economic circumstances and ways to improve them but also consider a broader range of indicators of what constitutes a good life, such as natural resources, safety, and social capital [8]. However, the definition and interpretation of a "good life" is often left up to policymakers [9].

Hence, the regeneration of these neighborhoods leads to far-reaching changes in their physical and social fabric. This raises questions of equity, sustainability, and (social) justice, as these processes are often followed by gentrification, which benefits new residents and disadvantages existing ones [10]. In this respect, citizen participation is seen as a crucial and undisputable factor for legitimizing regeneration projects and ensuring their successful implementation. Likewise, it is widely accepted that the regeneration of shrinking cities calls for new governance agreements between (local) governments and societal partners [11–14]. Involving residents in transformation of an area is not only seen as desirable but is also legislated by the Dutch government [15]: citizen participation is required if a regeneration project is to be initiated and conducted. In this regard, citizen participation is not only a functional requirement for the regeneration project but is also seen as a political act in which citizens are political actors and, as such, are involved in public decision making [16]. In general, however, this ideal of citizen participation is not put into practice, and the question remains as to how current participatory practices enable citizens to exercise influence as political actors in urban regeneration projects. In this paper, to develop an understanding of what political agency means, we examine this question based on Mouffe's [17] concept of the political, combining this with findings from our empirical research undertaken in Heerlen North.

In Section 2, we discuss (Section 2.1) territorial stigmatization and (Section 2.2) the current context of citizen participation in shrinking cities. Furthermore, we dwell on Mouffe's ideas "on the political" (Section 2.3) and the need for recognition practices (Section 2.4). We then discuss our methods, based on our qualitative research undertaken in Heerlen North (Section 3) and report the results and discuss them (Section 4). Finally, in Section 5, we offer our conclusions regarding how current practices of citizen participation impact the political agency of residents in areas undergoing urban regeneration.

## 2. The Problematic Nature of Urban Regeneration Projects: Territorial Stigma, Limited Citizen Participation, and Alternatives

### 2.1. Territorial Stigma as a Background for Regeneration Projects in Shrinking Cities

Territorial stigmatization has received limited attention in the urban regeneration discourse, and particularly in discussions of shrinking cities [4]. Nevertheless, it is an important phenomenon, as it determines the targeted areas' position in time and space and, as such, frames how they are interpreted and the policies and actions applied in their regeneration [18]. Many public anxieties about problematic neighborhoods can be observed in public and scientific debates. Represented with ongoing negative depictions in the media and in public discourse, such neighborhoods are denigrated and vilified [6]. Notions of the failure and undesirability of specific neighborhoods are omnipresent, and these places are often portrayed as symbolizing the margins of society [19]. Moreover, a poor physical layout is thought to cause increased social deprivation, decay, and criminality, which leads to concentrations of social problems and the exclusion of marginalized social groups in

these areas [20,21]. Wacquant [6], using Goffman [22] and Bourdieu [23] as starting points, analyzes the social stigma attached to living in these marginalized neighborhoods.

For Wacquant et al. [24], territorial stigma is the act of collectively representing a place in a harmful way. As such, it contributes to the social reproduction of "inequality and marginality in the city and, beyond [. . .]" (p. 1278). Territorial stigma is not merely a case of negative representation; it is produced [25]. Descriptions of corrupt and dystopian urban spaces frequently appear in Dutch political discourse [26,27]. For example, the Dutch Prime Minister Rutte spoke of "places that need to be reconquered from the scum of the streets" in a press conference in 2016, and Minister Dijkhoff claimed that certain urban places deserve exceptional treatment through the harsher sentencing of crimes. These externally imposed identities, which characterize certain places as disorganized or even dangerous, can result in public avoidance, selective out-migration, and disinvestment [28], which then leads to the further reproduction of the marginalization. These socio-spatial images are mobilized in the process of othering [29,30] and seriously impact the residents of these neighborhoods. These neighborhoods are thus imagined as different and disconnected from the city [24]. Moreover, territorial stigmatization implies a moral spatial order [6], labelling not only the places but also the people living in these places as morally inferior. This not only generates the expectation that residents are likely to have a problematic social status but also that they themselves are likely to develop deviant behavior. Hence, people in these places are discursively constructed as either lazy, victims, or criminals [6]. This understanding of the residents of these areas neglects the many ways in which they show resourcefulness and agency in shaping their environments—even under the most dire conditions, such as shrinkage [14].

Territorial stigmatization is a fact of life for many deprived neighborhoods, but particularly so for shrinking cities. That is, shrinking cities often suffer from a bad image [4,14,31]. Decay, poverty, and high crime rates are just some of the issues associated with shrinking cities as they are seen as losers in the race toward globalization. However, many shrinking cities contest these negative images [4]. Research shows that shrinkage does not necessarily lead to poor life satisfaction [4,14,32]. Nevertheless, stigmatization can trigger drastic redevelopment measures, which can disadvantage the residents [4].

It is important to understand this phenomenon, as it can frame approaches to regeneration and accompany citizen participation in regeneration projects.

## 2.2. Citizen Participation in Shrinking Cities and Urban Regeneration

A consistent policy response to the consequences of shrinkage and urban decay has been to reshape or demolish physical spaces with the expectation that social problems would diminish [33]. These processes of restructuring have been criticized, as they come with a high price: the displacement of people, the destruction of existing social networks, and gentrification [10,34]. Nevertheless, physical and socioeconomic restructuring is an important element in improving quality of life in these neighborhoods. One way to temper the possible negative outcomes of regeneration is ensuring citizen participation in shaping and conducting urban interventions. Involving citizens in regeneration projects is seen as desirable and has received considerable academic attention (see, for example, [31,35,36]). In the words of Arnstein ([37], p. 216): "The idea of citizen participation is a little like eating spinach: no one is against it in principle because it is good for you". Arnstein's famous words illustrate the normative, mostly positive connotations linked to citizen participation. This connotation is also present in The Netherlands, as citizen participation is not merely considered relevant by various actors (particularly policymakers), but it is mandatory [15]. Combined with a general climate of austerity and structural budget cuts, regeneration has led local governments to search for solutions in which residents take and hold more responsibility in addressing local challenges to livability [38]. Involving citizens in regeneration projects is assumed to be desirable for sustainable development [14,31], as citizens can help develop solutions for declining neighborhoods [39].

However, there is a world of difference between theory and practice. Current participatory practices of citizens in regeneration projects have been researched extensively and have attracted criticism.

Current citizen participation practices have raised many questions and prompted some criticism. In brief, the most well-known critique is that citizens are expected to follow municipal rules that are often very complex, which makes it difficult for them to exert influence [14,40]. Moreover, low levels of institutional trust in deprived neighborhoods limit citizen participation [14,39]. In addition, regeneration projects are embedded in unequal power relationships, and the issue of power in these projects is described as challenging [38]. Similarly, participants in the project often perceive power to be concentrated and held by the municipality [11,38,39,41]. Likewise, Ubels [39] argues that local government willingness to share decision-making power with residents and to reflect this in the municipal organizational structure and working routines depends on political will and on the preparedness and relevant skills of the civil servants involved. Furthermore, the extent to which residents are truly included in urban regeneration projects depends on the combination of resources (economic, social, and cultural capital) on which they can draw and the choices that are open to them within the norms and rules of the setting (symbolic capital) [18,41]. Consequently, diversity in neighborhoods is not necessarily reflected in diversity among the project participants [11,38]. It should be noted that these projects do not attract many participants from diverse backgrounds and such participants are thus underrepresented. Redevelopment policies, however, can exacerbate social and spatial inequities if explicit efforts are not made to include diverse residents [42]. This lack of diversity can pose a challenge to the realization of livability for all [43], along with the sustainability of the project's results. While citizens are asked to use their social capital to maintain the livability of the area [14], it can be argued that they lack other forms of capital that are not shared with them (in particular, symbolic and cultural capital).

Despite the multitude of arrangements that aim to give citizens a voice, well-known obstacles, such as those mentioned above, prevent residents from exerting a real influence in urban regeneration projects. In our view, none of these obstacles operate in isolation; rather, they arise due to the fundamental organization of democratic processes in our society, in which political issues are moralized, leaving stigmatized citizens with hardly any political influence. To analyze this issue in greater depth, we explore Chantal Mouffe's ideas "on the political" [17].

### 2.3. On the Political: Towards Citizen Participation in Shrinking Cities

Mouffe is well-known for her statement that conflict should be at the heart of democracy, whereas, in practice, most Western democracies are driven by consensus. The Netherlands might even be an exemplary case of this model, as the Dutch are famous (and perhaps notorious) for the invention of the 'Poldermodel'—a method of consensus-driven decision making that was dominant in economic and social politics and policymaking during the 1980s and 1990s. Central to such consensus-driven approaches to democracy is the assumption that general agreement about the organization of society and its institutions is possible. Based on a reasonable and rational exchange of arguments, in which every citizen is given the opportunity to participate, a society can develop overarching principles for good and just ways of living together, and it is in the light of such overarching principles that compromise becomes possible. Indeed, the well-known ideas of Habermas [44] on communicative rationality can be recognized in this description of consensus-based democracy.

Mouffe's counterarguments against this approach toward democracy essentially boil down to the statement that the political and the social are inseparable realms of life. Citizens occupy different positions in society and, consequently, have different possibilities and resources, for example, in terms of the social, economic, and cultural capital referred to above (in Section 2.2). Despite all of the good intentions and motivations to ground democratic institutions (or, on a more practical level, the living conditions in urban areas) in a rational dialogue, we need to acknowledge that such a dialogue is impossible due to

the real-life differences between citizens in terms of labor conditions, economic resources, education, and the social support system, among other factors. Hence, a dialogue in the political realm cannot be detached from the concrete, practical inequalities experienced in the social realm. This is due to the power imbalances and practical obstacles that result from real-life inequalities, which hinder an equal dialogue between stakeholders in a specific context (e.g., the barriers referred to in Section 2.2). However, solving the issue by removing formal barriers to participatory processes will not solve the issue, as, first and foremost, it is precisely because of social inequality that people differ fundamentally in their views on 'the good life' and on the importance and meaning of issues regarding health, sustainability, justice, and the built environment. Hence, citizens hold positions on democratic themes and institutions that are sometimes diametrically opposed and cannot be reconciled or combined. Whereas theories of communicative rationality such as that proposed by Habermas [44] assume that rational arguments might solve this problem, Mouffe is clear: a general consensus cannot be grounded in rationality, because it is precisely the *affective* commitment of citizens to fight for a better life that is part and parcel of political participation. A somewhat simplified yet illustrative example related to urban regeneration is the concept of place attachment: the emotional attachment of a person to a specific place, which is often based on personal experiences [14,45]. This attachment can be deeply felt and make a certain place worthwhile for people, contrary to rational arguments about sustainability or health. It is exactly this emotional meaning of a place that might motivate residents to fight for it. In consensus-based policy, however, there is little room for such considerations and for positions that go against the general consensus and are not motivated by rational arguments. In fact, viewpoints that challenge hegemonic views are often dismissed as immoral. According to Mouffe, "the political" is often enlisted in this moral register. A good example of this phenomenon is the recent discussion regarding the energy transition in The Netherlands. Tenants of poorly isolated housing pay the highest prices to warm their houses but lack the resources to improve their situations. Hence, they complained about gas prices and Dutch foreign policy. As a response, politicians framed high gas prices due to the war in Ukraine as "the price we should pay for democracy and international solidarity". In doing so, they questioned the solidarity of citizens who are struggling to make ends meet, instead of taking these worries seriously. In relation to several difficult issues faced by modern society, such implicit and explicit moral judgments are at work, which makes it impossible for true ideological and political debate to take place. This is especially the case in areas that suffer from territorial stigma, the residents of which are pitied or blamed for their poor living conditions.

Consequently, the goal of citizen participation is not so much to foster an open dialogue about livable cities, but to create public support for major renovations that represent hegemonic views of the city and city life. Many regeneration projects, for example, consider it a given that everyone should be healthy and care about the environment, and they aim to arrange the living environment accordingly—but are health and climate really prioritized by all stakeholders? And, if consensus about such "design principles for urban renewal" is not the answer, what is? Mouffe's answer might come as something of a surprise: a conflict-based democracy instead of a consensus-based democracy. That is, Mouffe makes the case for agonistic spaces, spheres where there is room for a passionate and cutting-edge ideological conflict. For Mouffe [17], an agonistic space is "a communal symbolic space, in which adversaries are considered legitimate adversaries" (p. 60 in the Dutch translation).

If there are no such places where citizens can fight ideological battles with each other, they will seek out other outlets to do so. Conflicting interests and views are then expressed antagonistically and may even become a threat to the democratic rule of law. For example, the recent farmer protests in The Netherlands demonstrate the possible negative implications of a lack of agonistic space, as they go against the rule of law and contribute to societal polarization. Hence, some negative effects of a lack of true democratic debate could be the creation and/or further development of an atmosphere of distrust, an attitude of conflict, and, as a result, a reinforced division between groups in society and

the strengthening of political power over citizens. The lack of institutional trust of citizens, which is often mentioned as a cause of limited citizen participation, exemplifies this. For example, the famous paper by Rodríguez-Pose [46], titled "The revenge of the places that don't matter", argues that, if citizens feel distrust toward authorities, they will be reluctant to support their policies. This, in turn, can cause a revolt against the status quo through a wave of political populism [46].

Mouffe's theories have far-reaching consequences. When we no longer strive for consensus but enable a permanent debate over fundamentally different ideologies, this will also impact existing power structures. Not only will they change, but agonistic spaces require that power structures are permanently under debate. However, when we are serious about citizen participation in urban regeneration projects and other social transformations, it seems inevitable that existing power arrangements will be taken into account and made more "fluid". This is because, as Davies et al. [47] and other scholars (for example [48,49]) argue, new forms of collaboration between societal and political actors are not situated in a vacuum but are structured according to existing patterns of power and domination. Hence, for citizen participation to become a reality, it is necessary to question the collective and distributive aspects of the existing power arrangements. How could agonistic space come to be an alternative approach to citizen participation in the regeneration of shrinking cities?

*2.4. Agonistic Space and Citizen Participation in Shrinking Cities: The Need for Recognition*

Democracy, as expressed in agonistic spaces, requires listening with respect, taking opposing views seriously, and creating spaces for these views to be challenged. The need for agonistic spaces is all the more relevant in deprived neighborhoods where residents permanently struggle to survive. As we have illustrated, territorial stigma is often associated with these areas, and, certainly, actual problems are at the root of territorial stigma, as residents deal with issues such as poor housing, low wages, and insecure jobs or unemployment, sometimes combined with permanent life challenges such as chronic illness.

However, as noted above, this territorial stigma also includes a moral judgment. In accordance with Mouffe, we can understand why such stigma and moral judgment render true citizen participation impossible, but Mouffe does not provide a detailed account of agonistic spaces and how residents can become legitimate adversaries. It is clear, however, that an agonistic space requires participants to recognize each other as political actors. Thus, to understand how we can create agonistic spaces to enable citizen participation in urban regeneration projects, we need to consider recognition as an important condition. Recognition means, quite simply, to be seen and to be valued. The seminal work by Honneth [50] distinguishes three types of recognition, two of which are relevant to our analysis. The first is social esteem, which entails a judgment of the qualities and social 'worth' of individuals. In contemporary society, we can posit that there is a lack of respect, that is, the moral standards of, for example, symbolic capital (e.g., a person's education and employment status) are used to extend respect to the few and exclude large groups of people (also relevant in this respect is [51,52]).

In this respect, Lamont [53] speaks of recognition gaps that are growing, particularly in neoliberal societies. These gaps refer to disparities in worth and cultural membership between groups in society. However, people cannot simply be commanded to treat others with respect. Mutual recognition has to be negotiated and even requires a struggle, according Honneth [50]. Moreover, according to Honneth, a lack of social esteem results in a lack of self-esteem. This is especially relevant to residents of stigmatized areas, as they are vilified and seen as morally inferior due to their low socioeconomic status. Moreover, practices of territorial stigmatization and infantilization intensify the internalization of negative narratives, affecting residents' self-esteem and (negative) perceptions of fellow residents [6]. This aggravates residents' already precarious positions.

A second type of recognition is respect, which is seen as crucial for true citizen participation and the creation of agonistic spaces. Respect implies that citizens are recognized as autonomous human beings who are capable of autonomous self-determination based on

moral norms and are able to participate in democratic discussions. This involves a political component, that is, not being subject to external forces that have not been legitimized to exercise rule—in other words, it is a matter of being respected in one's autonomy as an independent being [54]. In this respect, Ročak [14] writes of empowerment in the context of the regeneration of shrinking cities, regarding the need for people to have a voice that is listened to, people being involved in the processes that affect them, and people who can themselves take action to initiate change. At the same time, we can observe normative expectations among the elites designing and running the regeneration projects [55]. The assumption that they know better, alongside the paternalistic approach of wanting to apply this knowledge to people living in stigmatized areas, does not show respect as such, and it infantilizes people living in stigmatized areas that are targeted by regeneration.

The question remains, however, as to how citizen participation has taken shape in Heerlen North, and whether the theories of Mouffe "on the political" might help us to understand the (lack of) political agency of its residents.

## 3. Methods

### 3.1. Research Questions and Context

This research was performed by a postdoctoral fellow (the first author). The main research question of her postdoctoral study was how social relations and connections in deprived urban areas influence urban regeneration projects. One of the sub-questions, which focuses specifically on citizen participation and is the point of departure for this paper, is as follows: how do existing participatory practices enable citizens to exercise influence as political actors in urban regeneration projects? This question is specifically related to the state of citizens' trust and participation in local institutions.

We performed our qualitative study in the shrinking Dutch city of Heerlen, more specifically, in the Heerlen North area. Heerlen is located in the southeast part of The Netherlands (Figure 1) and has around 86,000 residents. It is the most populated municipality in the Parkstad Limburg region [8], which is situated in the border region of The Netherlands, far from the conurbation area (with Amsterdam, Den Haag, Rotterdam, and Utrecht as its four largest cities), which is considered the economic, political, and cultural center of The Netherlands. Heerlen was formerly known for its prosperous mining industry and is now struggling with the aftermath of deindustrialization. Currently, Heerlen has a low socioeconomic status compared with the national average [56]. Moreover, the socioeconomic problems are concentrated in the large area of town referred to as Heerlen North (Heerlen Noord), where more than 60% of the city's households are located. In this area, residents have, on average, far fewer opportunities for a good income, a decent home, or to stay healthy than residents in the rest of The Netherlands [57].

To improve the quality of life in Heerlen, numerous regeneration projects have been implemented. Most recently, within the large-scale Heerlen North National Program (Nationaal Programma Heerlen Noord, NPHLN), residents, businesses, governments, and civil society organizations are working together to develop a better future for Heerlen North over the next 25 years [57]. What distinguishes this program from other regeneration efforts is its long-term orientation and its focus on "brede welvaart" or "widespread prosperity". This indicates a holistic approach, where the regeneration projects address physical and economic circumstances and ways to improve them, while also considering a broader range of indicators of what constitutes a "good life" [9], including, for example, indicators related to health and safety.

### 3.2. Research Strategy

In order to answer the main research question, a qualitative research strategy was employed, including a documentation study and interviews. A secondary analysis of official statistics [8], as well as the available local data, was used to better understand why Heerlen North became a designated area for urban regeneration. To achieve this goal, information on the local context and widespread prosperity indicators were collected from

statistical reports [58], historical documents [59,60], and local policy documentation [57,61]. Moreover, we interviewed two groups: residents of Heerlen North and professionals involved in the regeneration projects. We chose to interview these two groups to shed light on participatory practices and social and institutional relations, which then improved our understanding of the political agency of citizen participation in the regeneration of shrinking cities. We conducted a total of 18 interviews and 2 group interviews. These included 12 interviews with individual residents and 6 interviews with social workers and other (social) professionals and municipal staff who are involved in regeneration and citizen participation. The two group interviews involved only residents. The interviews were conducted over the course of 13 months (from January 2022 to February 2023).

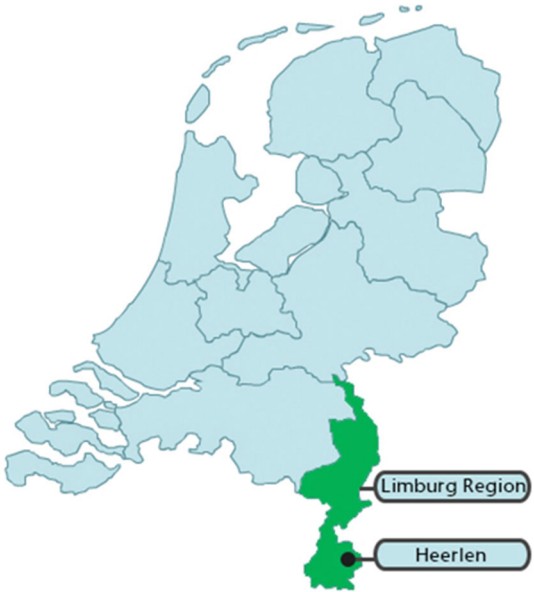

**Figure 1.** Location of Heerlen in The Netherlands.

*3.3. Selection of Respondents*

We used snowball sampling to find respondents for our project: professional and informal key informants with prominent roles in the regeneration projects were identified. These individuals are generally seen as community leaders. They were interviewed and/or directed us to other residents of Heerlen North, who were then asked for new respondents, et cetera. In this way, community leaders as well as individuals who were less active (at least in terms of citizen participation) were identified. The main data collection process (12 interviews and 2 group interviews) focused on this group. Although the resulting sample is not fully under control of the researchers when using snowball sampling, we aimed to reflect the demographic make-up of the city and explicitly searched for diversity during our selection of respondents.

*3.4. Data Collection and Analyses*

The interviews were semi-structured with an exploratory character. This offered the researchers the opportunity to examine respondents' experiences more deeply. All interviews were performed as informal conversations with pre-determined discussion themes, which were developed on the basis of the theoretical background of our study (widespread prosperity indicators, changes in the neighborhood, participation in regeneration projects, etc.). Moreover, we included demographic data (such as gender, age, education, and work experience); residents' views on the changes in the neighborhood; their experience of local problems (regarding different widespread prosperity indicators); their views on the community; their associations in the community; their motives for participation; issues of cooperation, trust, and accountability; power relations in the community; and informal

social networks and support. The interviews lasted from 1 to 1.5 h and took place in a suitable public place or at the interviewee's home. Four interviews were conducted using Microsoft Teams due to COVID-19 restrictions being in place at the start of the research project. All interviews were conducted by the same researcher. The analysis, conducted using ATLAS.ti software (ATLAS.ti Scientific Software Development GmbH, Berlin, Germany), followed four distinct steps: open coding, theoretical coding, selective coding, and integration with theory [62].

The research was performed in accordance with the Dutch Code of Conduct for Scientific Integrity [63]. We obtained informed consent from all interviewees, and all collected data were stored in a protected site, held by our university, as is customary according to the Dutch Code.

## 4. Findings and Discussion

*4.1. Professional View: The Existing Context of Citizen Participation as a Predictable but, in Some Respects, Insufficient Practice*

In this section, we present the interviews with the professionals involved in the regeneration projects.

The results indicate that citizen participation is currently organized via established groups and committees, usually represented by active citizens who are involved in numerous activities and embedded in many networks. These are engaged citizens of Heerlen North who are motivated to be involved in talks regarding the redevelopment of the area. In this respect, the picture that professionals paint resembles the well-known image of citizen participation described in Section 2.2. Respondents report a "well-oiled" machine of citizen participation. Procedures are known and the people involved know each other and know who to speak with if something needs to happen. This indicates that high levels of social, cultural, and symbolic capital are held by the individuals involved. That is, citizen participation practices are embedded in well-known procedures and language. For example, all respondents use the same professional terminology to describe practices and the goal they serve. In this respect, citizen participation practices are described and given meaning by professionals through the use of language that is heavily policy driven. However, it could be asserted that many of these practices use the language and forms of expression of the leading elites and, as such, limit other forms of expression. This results in a more elitist discussion, thereby limiting the political influence of residents who do not use these modes of expression. Multiple perspectives are far from prevalent. The same active citizens are often over-represented. These actors speak the language of policy and are rich in social capital. This puts those involved (both organizers and participants) at risk of overlooking fuzzy influences, power sharing, and social exclusion.

At the same time, the professional respondents acknowledge some of those risks, as they report the problem of citizens being represented by the "usual suspects". While the added value of citizen participation is acknowledged, the limitations of the same faces often attending participatory events (organized by, for example, the municipality) is recognized. These citizens have found a way to participate, often in municipal events; they build networks and speak (literally and figuratively) the language of "power". However, the call to include "common people" in regeneration processes is clear from the interviews with professionals involved in regeneration. Respondents report a desire to reach non-active residents and, at the same time, difficulty in doing so. "O well, people here have water on their lips here, are not concerned with talent development but with survival and you have to connect with that so it's about poverty and what you can do with that."

Professional respondents report that current participation opportunities are often about seeking consensus, which is seen as desirable. Moreover, they report being proud of their ability to fit into the existing system. They recognize that citizen participation practices are gatherings for like-minded people. This, in turn, results in the validation and legitimization of (for the most part) already-established citizen participation practices. This is astutely expressed in the following quote from a community worker: "we are aiming

to achieve harmonious coexistence". In this respect, practices of citizen participation are focused on sharing practical information and pedagogical actions but little or no social or political action.

Seeking consensus has another motivation: "not biting the hand that feeds you", as many organizations (and the professionals working in them) are dependent on municipal financing; they therefore report their reluctance to organize rebuttals to policies. Conversely, consensus seeking limits other forms of citizen participation. For example, Pløger [64] refers to Mouffe [17] and emphasizes the need to put "strife"—ongoing disputes over words, meaning, discourse, vision, or "the good life"—at the center of participatory processes. Striving for consensus by reducing decision making to technical issues, procedures, and expressions excludes other forms of expression and other groups who do not share the same form of expression, ignoring the political nature of regeneration projects. These consensus-guided procedures devalue particular styles of expression and the nature of the things expressed (for example, by non-active residents) while favoring others, especially those who use rational and reasonable speech. In this respect, Stapper and Duyvendak [55] speak of the distinction between "good" and "bad" residents. Although all residents are invited to participate, some entrepreneurial and well-spoken residents are more welcome than others. This leads to the configuration of moral categories in the citizen participation processes, which leads to the reproduction of social inequalities.

### 4.2. Citizen View: Participation Expressed in Feelings of Anger, Shame, and Fear

In this section, we reflect on the results of the individual and group interviews with citizens. Based on our analysis, several emotions related to citizen participation emerged: anger, fear, and shame. We present these emotions in relation to (self-)respect and (self-)esteem, conditions necessary for citizen participation in an agonistic space, as elaborated above. Further on, we discuss the implications of these results for agonistic space-based citizen participation in regeneration projects in shrinking cities.

#### 4.2.1. Anger Because of Experiences of a Lack of Recognition

Citizens report no knowledge about and no or very limited interest in participating in regeneration projects. These projects are experienced as being far from their own lifeworld. The respondents report other issues they must deal with in their everyday lives, such as paying bills, chronic health issues, or low-quality housing. Debt, and the daily practices connected to it, are experienced as dominating daily life and, as such, negatively impacting general health as well as social networks.

"But that's actually kind of my life, that's actually a guiding principle. That's always the guiding principle in my life. Debt problems, then your social quality deteriorates. So then you're actually just going to get a bit depressed about it. That's just a fact of life. That's not fun, huh. And then your social contacts also fall away, because you're basically always that girl who always has to say no to everything. And then you end up in a vicious circle."

As a consequence, this also negatively impacts citizen participation in regeneration projects. Having debt is connected with experiencing lack of agency and a fear of possible consequences. These issues impact daily life and managing them requires a lot of energy and resources. Being involved in regeneration projects is not seen as a priority or as desirable. This quotidian perspective is removed from the reality of the regeneration project. At the same time, respondents experience many problems in the neighborhood. In this respect, residents experience anger when reporting that their neighborhood is used as: "a garbage bin: we are not seen, not invested in"; "Heerlen North being used (...) like a sewage drain, so to speak". A lack of respect and esteem is experienced through disinvestment in housing and the physical environment of the neighborhood.

Furthermore, respondents report not receiving respect and esteem from administrators (these include civil servants and also social workers and other public professionals). Residents do not feel heard, seen, or taken seriously, which causes frustration.

"I just think they should take the people of Heerlen North seriously. That's the most important thing. We have to feel that they take us seriously and our problems in the neighborhood. That's the most important thing. Before you do anything as a municipality, go and ask people or something."

Moreover, being "a part of the system", by receiving a welfare benefit for example, means being limited in terms of agency, and having a reduced capacity to be the author of one's own life. The Dutch system of social support dictates strict rules about what recipients must do in order to receive a benefit: for example, apply for jobs, or not decline jobs. The whole system thus makes it clear that welfare recipients are considered lazy people who need financial incentives to make themselves useful to society [65]. Our data show that this misconception is explicitly felt by respondents. One respondent reported the "feeling that you constantly need to do as you're told" to be stigmatizing and unjust, limiting their agency. This, in turn, causes feelings of frustration and anger. These practices are experienced as disrespectful and cause humiliation (also described by [65]). Respondents are critical of back-to-work possibilities as they experience them as exploitative. The interviews are filled with the participants' frustration at not being able to exercise agency when it comes to these policies.

"And then you think okay, so Heerlen North is fucked again. We get to work for free again while that, at least that feeling is there.' 'If you really want to make sure that an area like Heerlen North is improving then you have to stop abusing (...) people and instead really wanting to help them and really wanting to make sure that things are improving."

This is experienced as unfair disciplinary action, which can be linked to the work of Wacquant [6] on disciplining the lower classes and Mouffe's [17] argument about the implicit and explicit moral judgments applied to unemployed citizens. In this respect, citizens report an unwillingness to be a part of participatory procedures put in place by the same municipality that decides whether or not one should continue receiving social support. Where there is little respect and a feeling of being morally judged is present, citizens are not seen as and do not see themselves as genuine political actors.

### 4.2.2. Fear as an Obstacle to Self-Esteem

The lack of agency explored above in Section 4.2.1 is also linked to the emotion of fear. This fear is related to the possibility of losing social support, thereby inducing the experience of insecurity. Moreover, the rigid structures of welfare arrangements leave citizens with little or no agency and a feeling of not being in control of their own lives. Citizens express a fear of institutional repercussions, such as a negative impact on their welfare benefit, if they raise their voice. This results in behaviors such as "not making too much noise" and staying invisible to professionals. This can be traced back to low trust in institutions due to experiences of a lack of respect and self-esteem. Insecurity and fear are also experienced with regard to accommodation. Having a house is seen as the most important thing in life. However, in the context of the Dutch social housing situation, others have the ability to decide whether and where a citizen should move; fundamental insecurity and precarity expressed through a lack of agency are thus just around the corner. This induces the fear that one might be moved to a different neighborhood far away from familiar networks and places.

Residents in this area experience another fear: the fear of criminality, and specifically criminal neighbors. In this respect, respondents report that they want to "stay under the radar". They do not want to be seen as snitches by their (criminal) neighbors, as this could put them in real danger:

> "But I'm not going to get involved in this no. I live here too. Should I take that into account huh? Before I also get a bomb in my home. Or have fire or whatever. I can listen and think along. But I can't talk along. No. Because I get punished here If I do that. So I don't."

Thus, they are not willing, for example, to attend any events that are organized by the municipality or engage in communication with anyone who could be seen as an authority.

Moreover, because of this fear of criminality, residents report the desire not to stand out too much, which is seen as a survival strategy.

In this respect, residents are limited by two types of fear: a fear of repercussions by the authorities, which is linked to basic needs (i.e., accommodation and social support), and a fear of criminality and repercussions from criminals if they are seen as snitches. These fears negatively impact residents' self-determination. In this environment, with its negative effects on self-determination (for example, the self-determination that would be exercised in an agonistic space), constructing an agonistic space is difficult if not impossible.

### 4.2.3. Shame as an Obstacle to Agonistic Space

Othering, stigmatization, and normative expectations are internalized and accompanied by feelings of shame. In turn, feelings of shame keep people quiet and prevent them from becoming involved in citizen participation events. In this respect, respondents express concern and shame about aspects of their lives that limit their participation. For example, their clothes come from clothing banks and are perceived as inferior. This limits the ways in which they choose to be involved in citizen participation events. Requesting that these citizens participate in a regeneration project acts as a form of cruel optimism [66]. While existing participation practices do not provide opportunities for individuals to express themselves in a way that is inclusive and not morally loaded, they nevertheless assume and promise the unachievable fantasies of the good citizens who participate in the redevelopment of the areas they live in.

Another factor that limits participation is the mode of expression of citizens. As discussed above, current practices of citizen participation use specific language and forms of expression. "I wouldn't understand it anyway. These meetings are for other kind of people". In the individual interviews and group interviews with citizens, we observed different forms of expression. For example, raised voices and tears from frustration. These means of expression are accompanied by the frustration of not being respected and not being able to practice self-determination. Citizens often express themselves differently, in terms of both style and content, from policy expectations and the style and content of the (power) elites. This form of expression is not valued by the existing power structures: for example, it is frowned upon to raise one's voice or not use (semi-)professional language.

When engaging in these less acceptable modes of expression, citizens are not taken seriously. They are seen as unreasonable and "bad residents" [55]. This could be seen as contributing to ornamental participation [67], which leads not to external exclusion but to internal exclusion [68], because there are indeed routes to citizen participation, but these are paved with invisible obstacles. However, according to Mouffe [17], it is specifically the *affective* commitment of citizens to fight for a better life that is part and parcel of political participation. The current situation is that citizen participation practices take place under conditions in which residents cannot express themselves properly, meaning that they are not fully recognized as political actors. Not knowing the right language with which to engage in discussions on regeneration projects is seen as a sign of inferiority. Residents feel that, even if they say something, it does not count. As such, although formally invited to take part in the debate, residents are not accepted as an equal political subject.

Finally, also related to shame is the perceived image of the neighborhood. Problems in the neighborhood and its bad image bother residents: "I would love to get in the car, drive away and never come back". This is linked to their experience of territorial stigma. Respondents report experiencing territorial stigma and being bothered by it in two ways: (1) it negatively impacts opportunities, for example, being discriminated against in the recruitment process because one comes from Heerlen North, and (2) being ashamed of coming from Heerlen North when interacting with outsiders. The otherness of the space and of the people living in it is constituted through its peripheral location at the edges of the city and the material landscape (in contrast with privileged areas) [6].

Agonistic space requires participants to recognize each other as political actors [17]. In the current context of citizen participation in regeneration projects, moral standards

of the "good life" and the decent citizen (the latter being linked to symbolic capital, including one's education and working status) are used to extend respect to the few and to exclude many [51–53]. Moral standards of what is good "for the people in the area" are applied to the policies and actions guiding regeneration projects. This, as a result, triggers anger, fear, and shame, which then negatively impact the opportunities for developing agonistic spaces.

## 5. Conclusions

Our research in Heerlen North demonstrates that, from the perspectives of professionals and residents, citizen participation is not an easy task. It is clear that citizen participation is flawed. This is not due to practical obstacles, such as communication about participatory events; rather, drawing on Mouffe, we can recognize the flaws in the organization of citizen participation as an expression of a certain (consensus-based) view of democratic decision making, which excludes citizens with different lifestyles and ideologies by moralizing them. This hinders true confrontations between differing views on the area and adequate urban interventions by citizens, professionals, and policymakers. In our analysis, a lack of recognition of local residents as a true political actors is part and parcel of the process of citizen participation. The existing institutions tend to view citizens from deprived areas as morally incapable of making sound decisions. This is humiliating and an act of disrespect. It also creates a discourse that makes it difficult for citizens to have their voices heard. In addition, our research shows that the day-to-day ordeals, challenges, and concerns of citizens remain far removed from the goals of urban regeneration. This makes it difficult for them to participate in urban regeneration projects; when in survival mode, participation in an urban regeneration project is not a priority. In addition, profound and consequential misconceptions arise when regeneration projects do not take into account the daily practices of the target group (or even overrule the group's concerns). This is because the basic requirement of feeling recognized is not met. Recognition means, first and foremost, to be seen, and to have one's ordeals and worries acknowledged.

While current participatory practices ask for input from residents, this does not make these people full-fledged political actors. Although urban regeneration projects have admirable goals, and we do not doubt that a sincere attempt is made to provide citizens with the opportunity to have an impact on the future of their neighborhoods, current participation practices unwillingly function as "ornaments of empowerment" [66] instead of being true vehicles of empowerment that enable the voices of "the powerless" to be heard. We hold that this is partly due to the implicit view of democratic practices at the heart of these projects; that is, they are based on a strong belief in consensus and the need for rational dialogue. Therefore, we call for future processes to include the voices and concerns of diverse populations [42]. This requires a movement toward the lifeworld of the residents of regeneration areas, which means a better understanding of their ordeals and concerns. Additionally, participatory practices must be open to otherness (in terms of language, as well as in terms of ideologies and normative orientations) and recognize citizens as political actors by showing them respect and esteem. Practices of territorial stigmatization imply that the neighborhood is the problem. Redevelopment should instead strive to generate respect and redirect resources from the buildings and appearance of a neighborhood toward their residents by creating the political, social, and material conditions that foster respect and esteem and, as such, urban social justice. More research should be undertaken to examine politicized participatory practices and how these take shape in order to understand and improve citizen participation in urban regeneration projects in shrinking cities.

**Author Contributions:** Conceptualization, M.R. and S.K.; methodology, M.R. and S.K.; software, M.R.; validation, M.R. and S.K.; formal analysis, M.R.; investigation, M.R. and S.K.; resources, M.R. and S.K.; data curation, M.R.; writing—original draft preparation, M.R.; writing—review and editing, M.R. and S.K.; project administration, M.R. All authors have read and agreed to the published version of the manuscript.

**Funding:** This research received external funding from Regieorgaan SIA.

**Institutional Review Board Statement:** Not applicable.

**Informed Consent Statement:** Informed consent was obtained from all subjects involved in the study.

**Data Availability Statement:** Not applicable.

**Conflicts of Interest:** The authors declare no conflict of interest.

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
