# Peer review of "The Eternal Struggle for the City: In Search of an Alternative Framework for Citizen Participation in Urban Regeneration Projects in Shrinking Cities"

_sustainability, doi:10.3390/su151612653_

Round 1
Reviewer 1 Report
1.After conducting a plagiarism check, excluding citations and references, the similarity score is 11%. According to the standards of SSCI journals, this similarity percentage is too high and requires appropriate revision.
2.The author's literature review covers many relevant sources; however, the presentation to readers appears as a disorganized and chaotic collection of references.
3.There is a lack of critical analysis of the literature and the author's synthesized perspective after comparing these sources. It is strongly recommended that the author improve this aspect; otherwise, the paper may fall short of being considered a scholarly work and resemble a mere book report.
4.While the author has made significant contributions to the study of phenomena such as "competitive spaces," there are still numerous contradictions and shortcomings. It is advisable for the author to refer to relevant publications by Neil Brenner and Soja on these topics.
5.The absence of geographical visualizations will hinder readers from imagining and comprehending the knowledge the author intends to convey.
6.The fundamental research design, necessary to adhere to scientific principles, is lacking. As a result, the author is unable to effectively communicate the knowledge they wish to share.
7.The absence of systematic knowledge presentation, organized and synthesized by the author, makes this writing style unsuitable for publication in an SSCI journal.
Author Response
Manuscript ID: sustainability-2507287
Type of manuscript: Article
Title: The eternal struggle for the city: In search of an alternative framework for citizen participation in urban regeneration projects in shrinking cities.
Sittard, 4th of August 2023
Dear reviewer,
Thank you for your valuable comments which we have used to rewrite our paper. This enabled us to further enhance the quality of our article. Below, we will explain, point by point, the details of the revisions to the manuscript and our responses to your comments. In attachment we’ve included the letter to the reviewer and the comments and our answers to the second reviewer. We hope we’ve succeed in this, and that our article is accepted for publication.
Thank you again for the opportunity to revise our paper.
Sincerely,
Maja Ročak & Sabrina Keinemans
- After conducting a plagiarism check, excluding citations and references, the similarity score is 11%. According to the standards of SSCI journals, this similarity percentage is too high and requires appropriate revision.
Author’s response: This paper is based on new research (we haven’t published in English about this study yet) and completely written by both the authors. We tried to address this comment by rearranging and reformulating the text (also as a response to comments 2 and 7), and we double checked and adjusted citations and references. Changes made are visible in the track changes version of the resubmitted paper. We hope this problem is solved by these actions.
- The author's literature review covers many relevant sources; however, the presentation to readers appears as a disorganized and chaotic collection of references.
Author’s response: We thoroughly rearranged and edited our literature review (Section 2), and also expunged several passages in an attempt to clearly focus on a specific item in every section. Also, we changed some of the headings and critically examined our use of concepts in order for the text to be more coherent and consistent. Please also see below our response to other comments where the changes we made are specified further.
- There is a lack of critical analysis of the literature and the author's synthesized perspective after comparing these sources. It is strongly recommended that the author improve this aspect; otherwise, the paper may fall short of being considered a scholarly work and resemble a mere book report.
Author’s response: We recognize that our analysis lacked a clear focus (also due to the comments of reviewer 2 about the main question we were trying to answer). What we were trying to do, is not so much critically discuss existing literature. Although several critical comments can be made about current literature on citizen participation and the work of Mouffe, this was not or goal, nor does the space of this paper does not allow to dwell on that. This comment, as well as comments of reviewer 2, did make us realize however, that the function of the theoretical notions in our paper was not clearly articulated and that the analysis did not logically indicate this. Therefore, we reformulated our central aim in the introduction of the article, and rewrote the theoretical sections in order to function as a (more) clear analytical lense. Please see section 2 for the changes we’ve made.
- While the author has made significant contributions to the study of phenomena such as "competitive spaces," there are still numerous contradictions and shortcomings. It is advisable for the author to refer to relevant publications by Neil Brenner and Soja on these topics.
Author’s response: The concept of competitive spaces is – although relevant to – not the focus of this paper, so we did not dwell any further on this concept in our current analyses. Our theoretical section was quite full with concepts and references: we revised this section and critically examined which concepts and authors were really relevant for our analysis and rearranged our argumentation in order to make it more coherent and avoid contradictions.
- The absence of geographical visualizations will hinder readers from imagining and comprehending the knowledge the author intends to convey.
Author’s response: We added a geographical map in order to visualize the – literally – marginal position of the city in the Netherlands (See section 3.1)
- The fundamental research design, necessary to adhere to scientific principles, is lacking. As a result, the author is unable to effectively communicate the knowledge they wish to share.
Author’s response: We completely revised the methodology section of our paper (Section 3). We added subsections which are in line with general outlines for scientific papers, in order to make sure that all relevant information about our design and research process was tackled. Were information was lacking (for example with regard to the research question) or unclear (for example with regard to the sampling process) we’ve added new information.
- The absence of systematic knowledge presentation, organized and synthesized by the author, makes this writing style unsuitable for publication in an SSCI journal.
Author’s response: With regard to this comment, we used the same strategy as in response to comment 1. We rearranged and edited all sections of our text, in order to organize it and make it more understandable and fit for publication in an SSCI journal.
Reviewer 2 Report
The manuscript is interesting - links the problem of citizen participation and shrinking cities. Section 1 Introduction and section 2 The problematic nature of urban regeneration projects: territorial stigma, consensus approaches and alternatives thoroughly describe the background of the issues raised, referring to the extensive literature. The theoretical part of the article is much more extensive than the practical part.
I would like to make a few corrections to the content of the article:
- "The paper answers the question: how could agonistic space be an alternative approach to citizen participation in the regeneration of shrinking cities?" the authors claim. In my opinion, the authors rather prove, based on interviews with residents, that agonistic space cannot be created under current conditions. My thought is also that, after all, the dispute and struggle (agonistic space) ends up developing a consensus on public space anyway. Isn't consensus a kind of democracy - we do what the majority wants.
- in the title of section 2.3. there is a reference to the literature and the citation page is given - this is rather not done in section titles
- in section 3 Method, line 378 the authors write that "Our main research method is a case study". A case study is not a method - a method can be tested/applied for a selected case (case stage). The methods used are described starting at line 398. The section also lacks information on how people were selected for interviews, (both residents and social professionals. It is also unclear how large the groups interviewed were (2 group interviews). Were these groups representative? There is no further information on the 12 individuals , who were interviewed. It is essential to complete this supplement.
Overall, the manuscript is a valuable study. Despite the comments, I believe that it deserves to be published in a revised version. I hope that the comments in the review will contribute to raising the scientific level of the manuscript.
I recommend its publication after the additions recommended in the review.
Author Response
Manuscript ID: sustainability-2507287
Type of manuscript: Article
Title: The eternal struggle for the city: In search of an alternative framework for citizen participation in urban regeneration projects in shrinking cities.
Sittard, 4th of August 2023
Dear reviewer,
Thank you for your valuable comments which we have used to rewrite our paper. This enabled us to further enhance the quality of our article. We hope we've succeeded in this, and that our article is accepted for publication. Below, we will explain, point by point, the details of the revisions to the manuscript and our responses to your comments. In attachment we’ve included the letter to the editor and the comments and our answers to the second reviewer.
Thank you again for the opportunity to revise our paper.
Sincerely,
Maja Ročak & Sabrina Keinemans
- The manuscript is interesting - links the problem of citizen participation and shrinking cities. Section 1 Introduction and section 2 The problematic nature of urban regeneration projects: territorial stigma, consensus approaches and alternatives thoroughly describe the background of the issues raised, referring to the extensive literature. The theoretical part of the article is much more extensive than the practical part.
Author’s response: Thanks for this overall judgment of our paper! We realize that the theoretical part of the article I quite lengthy, and although the reviewer does not urge us to shorten the text, we critically searched for abundant or recurrent passages and deleted them as much as possible (Section 2).
- I would like to make a few corrections to the content of the article:
"The paper answers the question: how could agonistic space be an alternative approach to citizen participation in the regeneration of shrinking cities?" the authors claim. In my opinion, the authors rather prove, based on interviews with residents, that agonistic space cannot be created under current conditions. My thought is also that, after all, the dispute and struggle (agonistic space) ends up developing a consensus on public space anyway. Isn't consensus a kind of democracy - we do what the majority wants.
Author’s response: This is very sharply noticed, and we do agree that the question ‘how could agonistic space be an alternative approach to citizen participation in the regeneration of shrinking cities’ is kind of rhetoric and actually does not adequately address the issue we want to discuss. What we aim to do, is analyze – with help of the ideas of Mouffe – whether citizen participation leads to true political participation, which enables residents to influence not only the ‘how’, but also the ‘why’ of urban regeneration projects. This comments makes us aware of the fact that this goal was not sufficiently expressed. Therefore, we reformulated the goal and central question of the article (see the track changes, Section 1 and Section 5), in order to address that question more precisely.
Further, as we conclude that a true political ‘conflict’ (in the words of Mouffe) has not yet been realized with regard to urban regeneration, we also cannot conclude that consensus is always the result. Also, consensus might be viewed as a kind of democracy, but there are alternative views as well. For example, Mouffe’s definition of democracy is not ‘what the majority wants’, but rather whether minorities are able to influence and exert power on political issues (and hence, might become the majority one day!). Our hypotheses is that we can rearrange and organize citizen participation in such a way, that this true conflict is realized (but agree that we do not have any proof of this yet).
- In the title of section 2.3. there is a reference to the literature and the citation page is given - this is rather not done in section titles.
Author’s response: We critically examined the section headings as a response to comments of reviewer 1, and also solved this issue by excluding this source and reformulating the section title to be more in line with the changes made
- In section 3 Method, line 378 the authors write that "Our main research method is a case study". A case study is not a method - a method can be tested/applied for a selected case (case stage). The methods used are described starting at line 398. The section also lacks information on how people were selected for interviews, (both residents and social professionals. It is also unclear how large the groups interviewed were (2 group interviews). Were these groups representative? There is no further information on the 12 individuals , who were interviewed. It is essential to complete this supplement.
Author’s response: We completely revised the methodology section of our paper (Section 3). We added subsections which are in line with general outlines for scientific papers, in order to make sure that all relevant information about our design and research process was tackled. Were information was lacking (for example with regard to the research question) or unclear (for example with regard to the sampling process) we’ve added new information.
- Overall, the manuscript is a valuable study. Despite the comments, I believe that it deserves to be published in a revised version. I hope that the comments in the review will contribute to raising the scientific level of the manuscript. I recommend its publication after the additions recommended in the review.
Author’s response: We hope that our modifications sufficiently address the issues both reviewers discerned in our paper, and we are convinced that the comments – and our adjustments – truly did improve the paper. We hope the reviewers and editors find this improved draft of our paper suitable for publication.
Round 2
Reviewer 1 Report
n the previous review, several significant revisions were suggested for this article, and the author has shown commendable effort in quickly addressing them while maintaining scientific rigor. The reviewers acknowledge that certain aspects have been largely improved, but there are still some sentences and errors in the English section that need attention. It is advised that the author thoroughly proofread the English text and consider seeking professional English editing assistance.
It is advised that the author thoroughly proofread the English text and consider seeking professional English editing assistance.
Author Response
Manuscript ID: sustainability-2507287
Type of manuscript: Article
Title: The eternal struggle for the city: in search of an alternative framework for citizen participation in urban regeneration projects in shrinking cities.
Sittard, 18th of August 2023
To the editors and the reviewer,
Herewith we would like to submit our revised paper entitled, ‘The eternal struggle for the city: in search of an alternative framework for citizen participation in urban regeneration projects in shrinking cities.’ To start with, we want to thank the reviewer for the kind comments and feedback. As advised we used professional English editing assistance and are resubmitting thoroughly edited paper. This enabled us to further enhance the quality of our paper. We hope we’ve succeed in this, and that our paper is accepted for publication. Thank you for the opportunity to revise our paper.
Sincerely,
Maja Ročak & Sabrina Keinemans
